# Psychosocial and Biological Outcomes of Immersive, Mindfulness-Based Treks in Nature for Groups of Young Adults and Caregivers Affected by Cancer: Results from a Single Arm Program Evaluation from 2016–2021

**DOI:** 10.3390/ijerph182312622

**Published:** 2021-11-30

**Authors:** David Victorson, Gretchen Doninger, Scott Victorson, Gwen Victorson, Lars Hall, Carly Maletich, Bradley R. Corr, Kathy Scortino, Zachary Burns, Lori Allen, Ian Rosa, Kelley Quirk, Adekunle Adegbemi, Johanna Strokoff, Kile Zuidema, Kelle Sajdak, Todd Mckibben, Angie Roberts, Thomas W. McDade, Amanda Boes, Katie McAlinden, Karen Arredondo, Christina Sauer, Kristin Smith, John M. Salsman

**Affiliations:** 1True North Treks, Evanston, IL 60201, USA; gretchen.doninger@gmail.com (G.D.); victorson.scott@gmail.com (S.V.); gwenvictorson@gmail.com (G.V.); carlymaletich@gmail.com (C.M.); bradcorr@gmail.com (B.R.C.); kathyscortino@truenorthtreks.org (K.S.); lallen1380@gmail.com (L.A.); ianmatthewrosa@gmail.com (I.R.); kelley.quirk@gmail.com (K.Q.); adegbemi@hotmail.com (A.A.); johanna.strokoff@gmail.com (J.S.); zuidemak@msu.edu (K.Z.); kksajdak@gmail.com (K.S.); mandyboes@truenorthtreks.org (A.B.); kikimac78@gmail.com (K.M.); karearr@yahoo.com (K.A.); christina.sauer@northwestern.edu (C.S.); kristinroederer@gmail.com (K.S.); jsalsman@wakehealth.edu (J.M.S.); 2Department of Medical Social Sciences, Northwestern University Feinberg School of Medicine, Chicago, IL 60611, USA; 3Robert H. Lurie Comprehensive Cancer Center of Northwestern University, Chicago, IL 60611, USA; 4Breakwater Expeditions, Sandpoint, ID 83864, USA; larshall7b@gmail.com (L.H.); todd@breakwaterexp.com (T.M.); angie@breakwaterexp.com (A.R.); 5Anschutz Medical Campus, University of Colorado Comprehensive Cancer Center, Aurora, CO 80045, USA; 6Encompass Learning, Duluth, MN 55804, USA; zach@encompasslearning.org; 7Department of Human Development and Family Studies, College of Health and Human Sciences, Colorado State University, Fort Collins, CO 80523, USA; 8Upper Penninsula Forestry Innovatoin Center, College of Agriculture and Natural Resources, Michigan State University, East Lansing, MI 48824, USA; 9Deparment of Anthropology and Institute for Policy Research, Weinberg College of Arts and Sciences, Northwestern University, Evanston, IL 60208, USA; t-mcdade@northwestern.edu; 10Wake Forest School of Medicine and Wake Forest Baptist Comprehensive Cancer Center, Winston-Salem, NC 27157, USA

**Keywords:** nature, connection, mindfulness, cancer, young adult, caregiver, peers

## Abstract

The COVID-19 pandemic has left many individuals suffering from “connection deficit disorder” given changes to the way we work, go to school, socialize, and engage in daily activities. Young adults affected by cancer between the ages of 18–39 have known this connection deficit long before the pandemic. Being diagnosed and treated for cancer during this time can significantly disrupt engagement in important educational, career, social, and reproductive pursuits, and contribute to increased stress, anxiety, depression, and other negative outcomes. Experiencing meaningful connection—with nature, with peers who understand, and with oneself, may help assuage this adverse effect of disconnect. A single arm within-subjects program evaluation was conducted to examine outcomes following participation in immersive, multi-night, mindfulness-based treks in nature in a sample of young adults (*n* = 157) and caregivers (*n* = 50) affected by cancer from 2016–2021. Pre to post-trek changes included significant (*p* < 0.001) self-reported improvements in feeling connected to nature (d = 0.93–0.95), peers (d = 1.1–1.3), and oneself (d = 0.57–1.5); significant (*p* < 0.001) improvements on PROMIS Anxiety (d = 0.62–0.78), Depression (d = 0.87–0.89), and Sleep Disturbance (d = 0.37–0.48) short forms; and significant (*p* < 0.05) changes in pro-inflammatory biomarkers (d = 0.55–0.82). Connection-promoting experiences like this have the potential to improve health and wellbeing in this population and serve as a model for others.

## 1. Introduction

The COVID-19 pandemic has left many individuals suffering from “connection deficit disorder” as a result of the collective separation, isolation, and social disconnection many have experienced through changes in the way we work, go to school, socialize, and engage in daily activities [1,2]. Young adults affected by cancer between the ages of 18–39 have been living the socially isolated “pandemic life” long before COVID-19 and often complain of feeling alone and separated from their peers and the life they once knew.

Being diagnosed and treated for cancer during this age period often leads to experiencing a host of disconnecting life experiences [3]. This may include quitting or postponing educational or professional pursuits; losing important friendships and relationships; facing challenges with dating or finding a romantic partner; feeling overly dependent on one’s parents; becoming infertile or having other reproductive health issues; experiencing a myriad of short and long-term treatment-related side effects; and suffering from significant depression, anxiety, and fear of recurrence [3,4,5,6,7,8]. As a result, young adults affected by cancer are at high risk of experiencing the “adverse effect of disconnect”. Experiencing meaningful connections may help lessen this burden and serve as an important source of support and resilience. 

Connecting with nature in different ways has received growing attention over the past several decades, both from the popular media, as well as from the scientific research community [9]. Mounting empirical evidence suggests that spending time in green and blue spaces can be good for our health and wellbeing [10,11,12,13,14,15,16,17]. While many questions remain regarding how we define nature contact and associated causal pathways and mechanisms of action, nature connection is increasingly considered an important health determinant for disease prevention and health promotion efforts [18]. 

Connecting with other beings in meaningful ways is another highly recognized form of connection with a considerable amount of supporting scientific evidence [19,20,21,22]. Compelling empirical data on the health benefits of feeling connected with others have been reported by researchers for decades for improved mood, decreased cardiovascular disease risk and even all-cause mortality risk [21,23,24]. Only recently have public health efforts begun to acknowledge and prioritize the importance of social connection in large-scale implementation initiatives. 

Connecting with oneself through mindfulness meditation is a more internally-focused type of connection also supported by decades of scientific investigation [25,26]. With Buddhist origins that stem back over 2500 years, mindfulness is characterized as connecting with our present moment experiences with an attitude of curiosity, openness, acceptance, and self-compassion [27]. Not unlike connection with nature and others, mindfulness has also been shown in a number of research trials to improve a host of emotional, cognitive, physical, and social domains, such as mood and negative affect [28,29], attention [30], and even regulation of biological processes associated with the body’s stress response [31,32]. 

Despite mounting empirical support for these connection-centered health determinants, they are often missing from conventional cancer treatment, survivorship settings and supportive care programs. In 2009, a cancer support nonprofit was established called True North Treks (TNT), whose mission is to support young adults and caregivers affected by cancer to “find direction through connection” through engaging in free, multi-night backpacking and canoeing treks in remote wilderness destinations. The purpose of this current report is to present pre-post program evaluation data from the past five seasons of TNT’s programming (2016–2021), which focuses on outcomes of connection, anxiety, depression, sleep disturbance, and biological inflammation.

## 2. Materials and Methods

### 2.1. Design and Setting

A single-arm, anonymous, pre-post program evaluation study design was implemented by TNT and approved by the Internal Review Board of Northwestern University (STU00215456) to analyze and disseminate these findings (see Figure 1). TNT has gathered anonymous, pre-post, participant-reported outcomes on its nature-based programs since its first trek in 2010. However, in 2016 they also began administering validated short forms of anxiety, depression, and sleep disturbance from the Patient Reported Outcomes Measurement Information System (PROMIS) [33,34] along with their standard survey questions. In addition, during the 2016 season, they collected finger-pricked dried blood spot samples from a subset of participants before and after treks to examine potential changes in levels of chronic inflammation. This report only includes data collected from 2016–2021 in which these outcomes were measured. Note, due to the COVID-19 pandemic, no treks were offered in 2020. 

To meet the diverse needs and interests of its young adult cancer survivor participants, TNT offers a variety of different ways to engage in its programming. Treks are either primarily backpacking or canoeing-focused and have been offered in different remote wilderness destinations, including Montana, Oregon, Wyoming, Idaho, Utah, Minnesota, the Upper Peninsula of Michigan, and the Bahamas. Treks are available either for survivors-only or for survivors who wish to bring a caregiver (defined as anyone who has “been there” for the survivor during their cancer journey). Trek lengths either include 5 nights of backcountry camping or three nights at a nature-based retreat facility located in a wilderness setting. 

### 2.2. Participants and Enrollment 

Eligible participants are either young adults between the ages of 18–39 who were diagnosed with cancer during this same age range or adult caregivers ≥ 18 years of age and may include a family member (e.g., parent, child, sibling, spouse/partner) or close friend. Interested survivors and caregivers submit an application found on TNT’s website. Survivors ask their primary cancer physician to complete an extensive medical form on their behalf, which is reviewed by TNT’s Medical Director who is a clinical oncologist. If the application is approved, participants communicate regularly with TNT personnel and trek guides to help them prepare for their experience, including what to expect and pack, as well as things they can do to be as mentally and physically prepared as possible. 

### 2.3. Trek Activities, Objectives, and Curriculum 

TNT’s programming is rooted in three crucial connections often missing from conventional cancer care, which include: (1) Connecting with nature after experiencing something as unnatural as cancer treatment, (2) Connecting with peers who “get it” and have walked a similar path, and (3) Connecting with oneself through mindful awareness practices or MAPs. While backpacking or canoeing in grizzly bear country may be considered adventurous to many, the purposeful inclusion of adventure-based activities (e.g., rock climbing, river rafting, surfing) is not generally how TNT engages the natural world. Rather, participants hike and paddle in nature at the pace of the slowest participants, being reminded that anywhere they are exactly where they are supposed to be. Nature is regarded as a containing, “holding environment” through which participants might be able to experience a new sense of physical and psychological openness, spaciousness, and embodiment; unbound by traditional conventions of the time, linearity, and modernity; an environment that models impermanence and transformation and is supportive of slowing down and pausing, reflecting and noticing, appreciating, taking inventory, and feeling grounded, re-centered and recalibrated. 

During the welcome session around the fire on the first night, participants introduce themselves and briefly share why they came, they learn about TNT’s history and mission, and they are reminded of TNT’s basic requests of them: (1) To be safe (e.g., wearing a PFD while swimming in a river or minding cliff edges when hiking), (2) To take care of themselves and their needs (e.g., asking for a second helping of food, telling a guide about a blister, bringing something up they wish to discuss and not waiting or expecting it will be brought up by others), (3) To actively look for opportunities to get each other’s backs (e.g., taking turns cooking or washing dishes, calling out a stump or branch on the trail, or just listening fully without interrupting or trying to offer advice); and (4) To lean into this unique experience because what they put in is directly proportional to what they will get back.

For many participants, this is their first time experiencing the outdoors in such a raw and immersive way, which can be both exciting and scary. TNT guides play a crucial role in establishing a safe, fun, and welcoming environment through which participants may encounter new opportunities to explore, reflect, learn and have fun. On TNT’s week-long backcountry treks there are usually three guides, two of whom are wilderness first responder (WFR) certified, and a third guide who has experience teaching mindfulness meditation and yoga. TNT’s nature retreat facility-based treks may or may not include WFR-certified guides, depending on whether overnight camping in the backcountry occurs. Safety and participant wellbeing are TNT’s first and foremost priorities and the guides help ensure that participants stay warm, dry, well hydrated, well-fed, and get a good night’s sleep. Participants are exposed to opportunities to build greater outdoor efficacy through learning skills, such as the ABC’s of packing a backpack, how to set up a tent, safely cooking outdoors, backcountry hygiene, plant and wildlife identification, tying knots, being “bear aware” (and aware of other potentially dangerous animals), paddling and steering a canoe, using a map, purifying water, learning primitive skills, etc. In addition to these foundational outdoor skills, TNT guides also model expedition attitude and behavior, including wilderness risk management, active communication, good judgment, planning and decision making, taking initiative, flexibility, and tolerance for adversity and ambiguity. One of TNT’s guiding values is the Nordic adage, “There is no bad weather—just bad gear”. Curating these outdoor survival skills and leadership attributes is akin to curating good “gear”, which can have direct and powerful applications to a person’s cancer survivorship toolkit and mindset. 

Each day, in addition to eating, hiking and/or canoeing, and having some un-programmed downtime, participants engage in MAPs, which include instruction in basic, experiential mindfulness practices that can help foster greater present moment awareness and self-compassion. Every morning at sunrise yoga a new daily mindfulness intention is introduced, which includes: Beginner’s Mind and Non-Judgment (Day 1), Accepting and Acknowledging (Day 2), Letting Go and Letting Be (Day 3), Self-Compassion and Lovingkindness (Day 4), and Gratitude (Day 5). Throughout the day this theme is informally explored in different ways through nature contact (e.g., during a silent hike or paddle, eating meditation, brief guided mindfulness practices and sit-spots), and each evening around the fire a more extended guided meditation is offered through practices focused on breath and body awareness, sensory awareness, open-monitoring, metta, etc. After each guided practice, participants are led through a period of mindful inquiry, in which they are taught to openly reflect upon and share different things they noticed and observed during their practice. Mindful inquiry can help participants gain insights and feel validated and supported that they are not alone in experiencing common challenges in this practice or think they are the only ones who are “not good” at meditating [35]. At times mindful inquiry may offer opportunities for participants to make observations and connections between what they noticed in their practice and their experience as a cancer survivor or caregiver. Lastly, woven throughout is extensive use of nature and mindfulness-relevant poetry to help reinforce and clarify themes. 

### 2.4. Data Collection Procedure 

All enrolled TNT participants completed an anonymous pre-trek assessment within one week of their trek, followed by engaging in their respective trek experience. Within one week of completing their trek, participants completed a similar post-trek assessment. Participants were asked to write the same 4-digit ID number they would remember on each assessment so their pre and post responses could be compared at a later time. The majority of assessments were completed using an online survey administration tool, however, some participants completed paper-pencil versions in which data were entered into a central database. 

Blood samples were collected from a subset of 2016 participants on the first and last afternoons of the trek (between 12–5 pm) by a trained TNT guide using standardized procedures [36]. After putting on surgical gloves, the guide sanitized the participants’ non-dominant ring finger with an alcohol pad. Next, a single-use disposable micro-lancet was used to prick the participants’ finger followed by gently pressing it from below the puncture site to allow a large drop of blood to form. The first drop of blood was wiped away with gauze, and the ensuing five blood drops were placed within each circle on the blood spot collection card. Samples were allowed to air dry completely before being placed in a large zip lock back and stored at room temperature until the trek was over at which time it was moved to a laboratory freezer at −30 °C. Samples were analyzed in duplicate batches in a laboratory led by co-author TM using high-sensitivity immunoassay protocols that have been validated to detect C-reactive protein (CRP) and Interleukin-6 (IL-6) from dried blood spot samples [37,38]. Average values were used in analyses. To minimize measurement error, all samples were analyzed in the same batch. 

### 2.5. Measurement of Outcomes 

#### 2.5.1. Connection

Participants answered three single-item indicators (written by TNT) at both time points related to: (1) Perceived feelings of connection to oneself and ability to reflect on one’s path and direction as a cancer survivor or caregiver, (2) Perceived feelings of connection with nature, and (3) Perceived feelings of connection with other young adult cancer survivors or caregivers. All items used a 5-point Likert response scale (Not at all, Very Little, Somewhat, Quite a Bit, a Great Deal).

#### 2.5.2. Knowledge and Efficacy

Participants answered three single-item indicators (written by TNT) at both time points related to: (1) Knowledge of the health benefits of mindfulness, (2) Confidence in one’s ability to incorporate mindfulness into daily life, and (3) Comfort doing outdoor activities like hiking, canoeing, backpacking, and camping. All items used a 5-point Likert response scale (Not at all, Very Little, Somewhat, Quite a Bit, a Great Deal). 

#### 2.5.3. Post-Trek Enjoyment, Appreciation, Insights, and Learning

Participants answered 15 single-item indicators (written by TNT) during the post-trek assessment related to enjoyment, appreciation, and things they learned as a result of their trek. All items used a 5-point Likert response scale (Not at all, Very Little, Somewhat, Quite a Bit, a Great Deal). The highest two categories of endorsement (e.g., “quite a bit” and “a great deal”) were presented as percentages. 

#### 2.5.4. Symptoms of Anxiety, Depression, and Sleep Disturbance

Participants answered validated questionnaires at both time points using the 4-item versions of the PROMIS Anxiety, Depression, and Sleep Disturbance short forms that have a 5-point Likert response scale (Never, Rarely, Sometimes, Often, Always, or Not at All, A Little Bit, Somewhat, Quite a Bit, Very Much). Anxiety items focused on feeling fearful, uneasy, overwhelmed by worries, and having difficulty focusing on anything besides anxiety; Depression items dealt with feeling worthless, helpless, hopeless, and depressed; and Sleep items pertained to having problems sleeping or falling asleep, how refreshing sleep was, and overall sleep quality. 

#### 2.5.5. Inflammation

Survivor (*n* = 16) participants from three 2016 week-long backcountry treks (Green River of Utah, Boundary Waters of Minnesota, Selkirk Mountains of Idaho) provided guide-collected whole dried blood spots to assess inflammatory markers IL-6 and CRP using a standardized minimally invasive finger prick technique. 

### 2.6. Statistical Analysis

Statistical analyses were conducted with IBM SPSS Statistics for Windows Version 27.0. Descriptive statistics were calculated for all variables using measures of central tendency and frequencies. PROMIS raw scores were converted into standard T scores (Mean = 50, Standard Deviation = 10) prior to analysis. Raw CRP and IL-6 estimates were logarithmically transformed using log10(x) prior to analysis to adjust for non-normal distributions. General linear models were applied to examine pre-post differences in outcomes in separate cancer survivor and caregiver analyses using *p* < 0.05 as a threshold of statistical significance. Exploratory comparisons examined the role of the primary mode of in-field locomotion (e.g., canoeing vs. backpacking focused) and trek type/duration (e.g., week-long backcountry vs. long-weekend nature retreat) on post-trek outcomes, holding pre-trek values constant. Cohen’s d effect sizes were calculated to determine the magnitude of effect, taking the correlation between pre and post-test values into account per recommendations from Morris [39] for within subjects estimates. Cohen [40] suggested small effects were between 0.2–0.4, medium effects between 0.5–0.07, and large effects ≥ 0.08. 

## 3. Results

### 3.1. Socio-Demographic and Cancer-Related Characteristics 

During 2016–2021, 157 young adult cancer survivors and 50 caregivers participated in treks (see Table 1). The majority of cancer survivors identified as female (75%) while the majority of caregivers identified as male (64%) with an average age of 33.6 and 39.1, respectively. Participants represented all major regions of the United States, including minor representation from Canada (*n* = 4) and Australia (*n* = 1). The majority of cancer types were breast (30.6%) and hematological (28.7%). During this time period, slightly more participants went on canoeing-focused treks (*n* = 91) compared with backpacking (*n* = 43). Nearly double went on week-long treks in the backcountry (*n* = 134) compared with long-weekend mini treks at a nature retreat facility (*n* = 73). These differences are likely more of a reflection of permits requiring smaller group sizes on backpacking treks and the mini-treks just starting in 2016 than other factors like participant preference. Complete, pre-post survey information was received from roughly 140 cancer survivors and 40 caregivers, a response rate > 80%. 

### 3.2. Changes in Connection 

Both survivors and caregivers had significant (*p* < 0.001) score increases in being able to reflect on their path and direction as a survivor or caregiver (*d* = 0.57 and 1.5, respectively), feeling connected to nature and the outdoors (*d* = 0.95 and 0.93, respectively), and feeling connected to other survivors or caregivers (*d* = 0.83 and 1.3, respectively). See Table 2.

### 3.3. Changes in Knowledge and Efficacy

Similarly, both survivors and caregivers reported significant (*p* < 0.001) increases in awareness of the health benefits of mindfulness (*d* = 0.89 and 1.4, respectively) and confidence in incorporating mindfulness into their daily lives (*d* = 1.3 and 1.5, respectively). Survivors, but not caregivers, had significant score increases (*p* < 0.001) in feeling comfortable doing outdoor activities. See Table 2.

### 3.4. Trek Enjoyment, Appreciation, Insights, and Learning

At the post-trek assessment, while the vast majority of survivors and caregivers reported enjoying their trek experience (89.3% and 92.5%, respectively), the caregiver rating (mean = 4.92/5.0) was statistically significantly higher (*p* = 0.02) than survivor rating (mean = 4.74/5.0). Similarly, while the majority of both groups reported intentions to spend time in the outdoors after they returned home (86.7% and 90%, respectively), the survivor rating (mean = 4.77/5.0) was statistically significantly higher (*p* = 0.04) than the caregiver rating (mean = 4.61/5.0). No other significant group differences were observed with the majority of survivors and caregivers reporting: (1) greater appreciation for nature and the outdoors (86.8% and 90%, respectively), (2) feeling like this experience will continue to affect them in positive ways after they return home (88.7% and 92.5%, respectively), (3) learning new skills, such as managing stress and uncertainty (74.1% and 87.5%, respectively), (4) being kind and compassionate to themselves (77.2% and 90%), (5) learning to be more accepting of themselves and their experiences (79.1% and 85%, respectively), and (6) planning to continue practicing mindfulness meditation (78.5% and 80%, respectively). See Table 3. 

### 3.5. Changes in Anxiety, Depression, Sleep Disturbance and Inflammation 

Survivors and caregivers both saw similarly significant (*p* < 0.001) decreases in symptoms of anxiety, depression, and sleep disturbance on PROMIS short forms. Biomarker analysis of a subset of 2016 survivors (*n* = 16) resulted in a moderate, significant (*p* < 0.05) decrease in IL-6, as well as a large, significant (*p* < 0.01) increase in CRP. See Figure 2 and Figure 3.

### 3.6. Primary Mode of In-Field Locomotion and Trek Type/Duration

A set of exploratory analyses were conducted to examine the role of the primary mode of locomotion during week-long backcountry treks (e.g., backpacking [*n* = 44 survivors; *n* = 8 caregivers] vs. canoeing [*n* = 68 survivors; *n* = 19 caregivers]) as well as trek type and duration (e.g., week-long backcountry treks [*n* = 94 survivors; *n* = 19 caregivers], vs. long-weekend nature retreats [*n* = 46 survivors; *n* = 19 caregivers) on outcomes of reflection, connection, knowledge, efficacy, and symptoms of anxiety, depression, and sleep disturbance. In a comparison of mode of locomotion, there were no significant differences (*p* > 0.05) on any of the outcomes. 

Similarly, in comparisons between trek type/duration there were no significant differences (*p* > 0.05) on the majority of outcomes between groups, except that survivors who participated on a week-long backcountry trek reported significantly higher (*p* = 0.03) increases in feeling connected to nature (post-trek mean [SD] = 4.7 [0.54]) compared with those on a long-weekend nature retreat (post-trek mean [SD] = 4.4 [0.86], ES = 0.45. Further, survivors and caregivers on a week-long backcountry treks reported significantly higher (*p* < 0.01) sleep disturbance scores (post-trek mean [SD] = 49.9 to 52.1 [8.2 to 9.8], respectively) compared with those on a long-weekend nature retreat (post-trek mean [SD] = 44.9 to 44.5 [9.3 to 8.2], respectively, ES = 0.58 to 0.85, respectively.

## 4. Discussion

Primary findings indicate that both cancer survivor and caregiver participants’ scores significantly improved in: (1) connecting to nature, (2) connecting to other survivors/caregivers, (3) connecting to oneself through reflecting on one’s path as a survivor/caregiver, (4) understanding the health benefits of mindfulness, and (5) feeling confident applying mindfulness to daily life, all with moderate to large effects. While only survivors had a small, significant score increase in feeling more comfortable doing outdoor activities after their trek, it should be noted that caregiver scores in this domain were already quite high at baseline and improved to roughly the same level as survivors at the follow-up assessment. The majority of participants reported enjoying, appreciating, and learning numerous things related to nature, stress management, mindfulness, and health behaviors. 

Survivor and caregiver groups both saw significant and meaningful reductions in symptoms of anxiety, depression, and sleep disturbance, representing minimally important differences in each domain [41]. Of note, the average baseline anxiety scores of survivors and caregivers placed them in the low moderate to high mild level of severity (T = 60.4 and 57.8, respectively), which decreased to within normal limits at the follow-up assessment, a clinically meaningful reduction. While meaningful score decreases were also observed in depression and sleep disturbance scales, these scores were all within a sub-clinical range of severity for these domains. 

A significant, moderate decrease was seen in IL-6 values between pre and post-trek, as well as a significant, large increase in CRP values during this time period. A closer inspection of un-transformed estimates reveals that mean pre-IL-6 values (16.7 pg/mL) were within the mild to moderate range of severity and decreased to within normal limits at the post-test (2.1 pg/mL) [42], which is a clinically meaningful reduction. For CRP, both pre and post values (1.03 mg/dL and 1.60 mg/dL, respectively) were within the normal range [43]. While it may have been expected that both biomarkers rise and fall in unison (suggesting the possibility of a distinctive mechanism of concurrent inflammatory activity), given that CRP is a good indication of muscle inflammation, this slight rise (staying within normal values) may be explained by the physical activity of the treks [44]. Given known associations between symptoms of depression and anxiety and acute inflammation [45,46] and the secretion of pro-inflammatory cytokines [47,48], it is important to advance our understanding of how immersive experiences in nature might “get under the skin”. This said, the evaluation design and very small sample of these biomarker analyses place these findings at the very preliminary level of understanding, which should be considered when interpreting these results. 

No significant differences in outcomes were observed between canoeing and backpacking-focused week-long treks in the backcountry. A moderate, significant difference was observed, however, in feeling more connected to nature among participants who spent a week in the backcountry compared with those on a long-weekend nature retreat. This is an interesting and somewhat logical finding; that camping, hiking, and canoeing in the backcountry for five nights may lead to greater appraisals of nature connectedness than staying at a nature retreat facility with only day hikes and day paddles. However, it should be noted that nature connection scores in the retreat group were also comparatively high at follow-up (Mean = 4.4, compared to Mean = 4.7 in the backcountry group), which should be considered when interpreting this finding. Finally, a significant difference in sleep disturbance was observed between groups, whereby participants on week-long treks in the backcountry reported relatively poorer sleep experiences compared to long-weekend nature retreat participants. This too is not surprising, that sleeping indoors in a bed (with access to an indoor bathroom) for a shorter period of time may lead to more favorable sleep reports. For context, however, it should be noted that for both groups, average post-trek sleep scores were both within normal limits. 

Previous research has supported many of the findings from this evaluation, such as increased reports of connection to the natural world and associated feelings of closeness to others [49], improved mood and sense of wellbeing [13], and increased self-efficacy [50]. While biological data have been collected in the context of brief, Shinrin-Yoku (forest bathing) studies [51], to our knowledge, this is the first evaluation to collect finger-pricked whole blood spots in the backcountry wilderness for the purpose of examining changes in pro-inflammatory biomarkers, such as IL-6 and CRP. This proved to be a safe, feasible, and convenient field-friendly method that has the potential to expand the measurement toolkit of nature and health researchers so that emerging biological outcomes may be included in the understanding of nature’s role on bio-behavioral determinants and mind-body processes. 

While this program evaluation helps advance knowledge of the potential impact of immersive, mindfulness-based treks in nature for groups of young adults and caregivers affected by cancer, these findings should be considered within the context of their limitations. First, the single-arm, within-subjects design, while appropriate for program evaluation purposes, prohibits any causal inference that changes in outcomes resulted directly from trek participation and may have been influenced by other unrelated factors and forces. Due to the anonymous nature of the data collection method, it was not possible to examine the influence of other important contextual factors on outcomes, such as certain socio-demographic or cancer-related variables. Further, the relatively lower levels of representation of male survivors and people of color in this sample (25% and 18%, respectively) place restrictions on the generalizability of findings. While it has been reported that cancer support services for young adults are less utilized by men and people of color, these estimates are often confounded by a lack of awareness regarding the existence or availability of such services [52,53]. Further, simply being aware of a service doesn’t always translate into actual use [53]. Future nature programming initiatives, such as this will be strengthened by implementing more rigorous evaluation designs and enacting purposeful strategies to increase representation and reduce inequities to nature access.

## 5. Conclusions

Overall, psychosocial and biological outcomes improved after participation in immersive, mindfulness-based treks in nature in a sample of young adults and caregivers affected by cancer. Continued implementation and study of immersive, mindfulness-based nature programs with this population are warranted as it holds the potential to increase access to age-appropriate, supportive care opportunities that improve quality of life and well-being. It may also serve as an exemplar model that could be replicated with other populations that experience the adverse effect of disconnect. 

## Figures and Tables

**Figure 1 ijerph-18-12622-f001:**
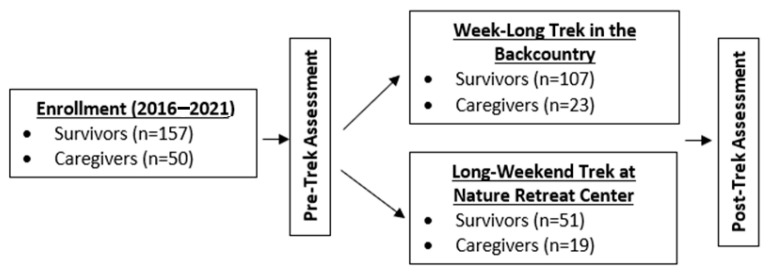
Study Flow.

**Figure 2 ijerph-18-12622-f002:**
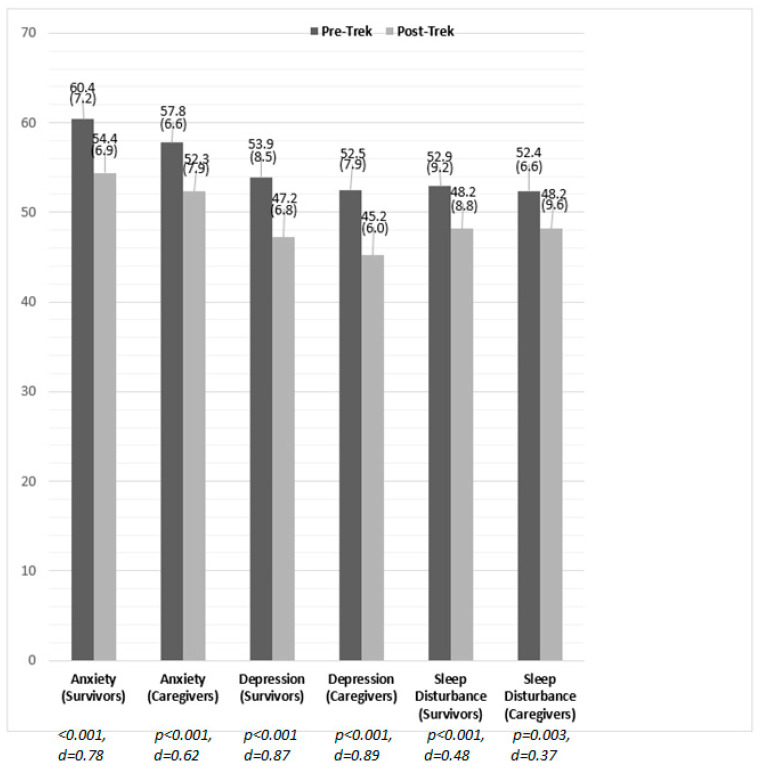
Pre-Post Changes in PROMIS Anxiety, Depression and Sleep Disturbance among Survivors (*n* = 139–141) and Caregivers (*n* = 37–38), 2016–2021. Note: Significance values and effect sizes represent pre-post changes within each respective group, and not comparisons between survivors and caregivers.

**Figure 3 ijerph-18-12622-f003:**
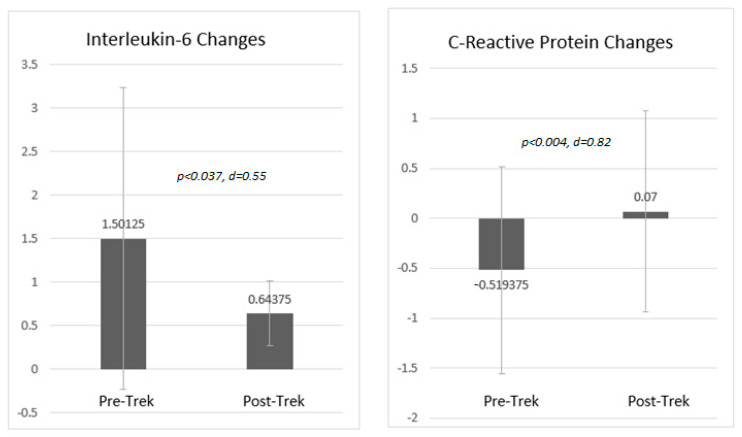
Pre-Post Changes in Interleukin-6 and C-Reactive Protein among Survivors (*n* = 16), 2016 Note: Values reflect log-transformed estimates.

**Table 1 ijerph-18-12622-t001:** Socio-Demographic and Cancer-Related Characteristics of Participants (2016–2021).

	All (*n* = 207)		Cancer Survivor (*n* = 157)		Caregiver (*n* = 50)	
	Mean (SD)					
Age	34.9 (7.4)		33.6 (5.3)		39.1 (10.7)	
Age at first cancer diagnosis	--		26.6 (8.2)		--	
Years since first cancer diagnosis	--		6.7 (7.1)		--	
	Value (Percentage)					
Sex						
Female	157	(76.0%)	117	(75.0%)	18	(36.0%)
Male	50	(24.0%)	40	(25.0%)	32	(64.0%)
Race						
White	163	(82%)	--	--	--	--
Non-White	35	(18%)	--	--	--	--
US Region						
Northeast	42	(20.8%)	--	--	--	--
Southeast	29	(14.4%)	--	--	--	--
Midwest	86	(42.6%)	--	--	--	--
Southwest	9	(4.5%)	--	--	--	--
West	36	(17.8%)	--	--	--	--
Primary Cancer Type						
Brain	--	--	19	(12.1%)	--	--
Breast	--	--	48	(30.6%)	--	--
Colorectal	--	--	6	(3.8%)	--	--
Gynecologic	--	--	3	(5.7%)	--	--
Head and Neck	--	--	4	(2.5%)	--	--
Hematological	--	--	45	(28.7%)	--	--
Kidney & Renal	--	--	1	(0.6%)	--	--
Melanoma	--	--	2	(1.3%)	--	--
Mesothelioma	--	--	1	(0.6%)	--	--
Sarcoma	--	--	12	(7.6%)	--	--
Stomach	--	--	2	(1.3%)	--	--
Testicular	--	--	3	(1.9%)	--	--
Thyroid	--	--	5	(3.2%)	--	--
Trek Locations from 2016–2021						
Bahamas	10	(4.8%)	5	(3.2%)	5	(10.0%)
Boundary Waters (Minnesota)	21	(10.1%)	21	(13.4%)	0	(0.0%)
Green River (Utah)	60	(29.0%)	40	(25.5%)	20	(40.0%)
Selkirk Mountains (Idaho)	19	(9.2%)	14	(8.9%)	5	(10.0%)
Upper Peninsula (Michigan)	63	(30.4%)	47	(29.9%)	16	(32.0%)
Wind Rivers (Wyoming)	23	(11.1%)	19	(12.1%)	4	(8.0%)
Yellowstone (Wyoming)	11	(5.3%)	11	(7.0%)	0	(0.00%)

Note: Missing values represent types of data that are only relevant to one group (e.g., cancer survivors, but not the other (e.g., caregivers).

**Table 2 ijerph-18-12622-t002:** Pre-Post Changes in Reflection, Connection, Knowledge, and Efficacy (2016–2021).

	Cancer Survivors (*n* = 140)						Caregivers (*n* = 38)					
	Pre-Trek		Post-Trek		Sig	ES	Pre-Trek		Post-Trek		Sig	ES
	M	SD	M	SD			M	SD	M	SD		
**Connection Item Indicators**												
Feel connected to nature and the outdoors	3.6	0.99	4.6	0.66	0.001	0.95	3.6	0.97	4.5	0.74	0.001	0.93
Feel connected to other young adult cancer survivors	2.9	1.1	4.4	0.83	0.001	1.1						
Feel connected to other caregivers affected by cancer	--	--	--	--	--	--	2.5	1.1	4.3	0.89	0.001	1.3
Able to reflect on my path & direction as a survivor	3.6	0.96	4.3	0.80	0.001	0.57	--	--	--	--	--	--
Able to reflect on my path & direction as a caregiver	--	--	--	--	--	--	3.2	0.89	4.5	0.64	0.001	1.5
**Knowledge and Efficacy Item Indicators**												
Aware of benefits of mindfulness for health	3.4	1.1	4.5	0.75	0.001	0.89	2.9	1.2	4.3	0.68	0.001	1.4
Comfortable doing outdoor activities	3.9	0.96	4.3	0.80	0.001	0.37	4.1	0.97	4.2	0.85	0.399	0.18
Confident incorporating mindfulness into daily life	2.8	1.0	4.4	0.84	0.001	1.3	2.7	0.91	4.2	0.71	0.001	1.5

Note: M = Mean; SD = Standard Deviation; Sig = Significance; ES = Cohen’s d effect size; Response Options of Single Items: 1 = Not at all, 2 = Very little, 3 = Somewhat, 4 = Quite a bit, 5 = A great deal; Missing values represent types of data that are only relevant to one group (e.g., cancer survivors, but not the other (e.g., caregivers).

**Table 3 ijerph-18-12622-t003:** Percentage of Post-Trek Enjoyment, Appreciation, Insights, and Learning (2016–2021).

	Cancer Survivors (*n* = 146)			Caregivers (*n* = 40)			
Single Item Indicators	Quite a Bit	Great Deal	Total	Quite a Bit	Great Deal	Total	*p*
Had fun.	22.8	66.5	89.3	7.5	85.0	92.5	0.02
Appreciation for nature and the outdoors increased.	26.0	60.8	86.8	22.5	67.5	90.0	0.69
Gained insights into things that can cause stress, frustration or discomfort.	32.3	39.9	72.2	42.5	42.5	85.0	0.59
Learned some things to help manage stress and uncertainties in life.	32.3	41.8	74.1	52.5	35.0	87.5	0.09
Learned ways to slow down and just notice mind and body.	25.3	53.2	78.5	27.5	57.5	85.0	0.99
Learned different ways to “respond” to stress instead of “reacting” to it.	27.2	39.2	66.4	37.5	30.0	67.5	0.18
Learned about being a more accepting “observer” to myself and my experiences.	36.1	43.0	79.1	42.5	42.5	85.0	0.65
I learned about being kind and compassionate to myself, even for the “little things.”	29.1	48.1	77.2	35.0	55.0	90.0	0.94
Gained deeper appreciation for some of life’s simpler things (e.g., walking, eating)	30.4	52.5	82.9	15.0	62.5	77.5	0.07
Learned about “sitting” with unpleasant experiences without becoming overwhelmed.	31.6	36.1	67.7	37.5	40.0	77.5	0.87
Have a better understanding of what mindfulness is and isn’t.	38.6	43.7	82.3	55.0	35.0	90.0	0.13
Feel more confident in ability to do things to stay healthy and well as a cancer survivor.	37.3	37.3	74.6	38.5	41.0	79.5	0.87
Plan to keep learning and practicing mindfulness meditation when return home.	28.5	50.0	78.5	40.0	40.0	80.0	0.16
Plan to spend more time in the outdoors, even if it’s just at a park, after return home.	19.6	67.1	86.7	35.0	55.0	90.0	0.04
Feel like this experience will continue to affect me in positive ways after return home.	16.5	72.2	88.7	7.5	85.0	92.5	0.13

Note: Response options were 1 = Not at all, 2 = Very little, 3 = Somewhat, 4 = Quite a bit, 5 = A great deal; *p* = significance level from mean comparison.

## Data Availability

Data used in this program evaluation will be made available upon request.

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
