# Peer review of "Psychosocial and Biological Outcomes of Immersive, Mindfulness-Based Treks in Nature for Groups of Young Adults and Caregivers Affected by Cancer: Results from a Single Arm Program Evaluation from 2016–2021"

_ijerph, 2021, doi:10.3390/ijerph182312622_

Round 1
Reviewer 1 Report
The authors present a novel, rigorous quasi-experimental study with clear objectives that contribute to the generation of evidence of the effects of the practice of Mindfulness in cancer caregivers and survivors.
The differential outcome between caregivers and cancer survivors is a valuable effort to describe their psychosocial functioning. The use of inflammatory biomarkers in this type of study is of utmost importance to establish the usefulness on the health (physical and emotional) of both groups. It is possible that the realization of the same Mindfulness practice has improved the social connection between the caregiver and the patient and, therefore, the interaction between the biopsychosocial components.
Here are some points to consider to enrich your study: It is convenient to make a diagram of the research design.
2. The presentation of results is not clear. Table 3 requires adjustments: 1) Remove the% symbol and mention it in the table title. 2) I recommend an X2 or ratio analysis to determine if there are significant differences between the two groups.
3. Figure 1 and Figure 2 do not present units of measure on the Y-axis.
4. In figure 1 it is not clear if the p-value and Cohen's d is the difference between the groups or between the pre and post-TREK? (The figure should be self-explanatory).
5. In figure 2, the hardest and most interesting data of the study, it is convenient to place error bars.
6. In the discussion, it is possible that the decrease in IL-6 and CRP concur, as explained by the authors. Would this result be the expected one to replicate the study or a similar investigation? If this is the case, the possibility of a distinctive mechanism of concurrent inflammatory activity should be suggested for the practice of Mindfulness.
Author Response
We thank Reviewer 1 for these very helpful suggestions. Please find our responses underneath each comment.
- It is convenient to make a diagram of the research design.
We have created a diagram of the research design (now Figure 1)
2. The presentation of results is not clear. Table 3 requires adjustments: 1) Remove the% symbol and mention it in the table title. 2) I recommend an X2 or ratio analysis to determine if there are significant differences between the two groups.
We have removed the % symbol, mentioned it in the title, compared survivor and caregiver percentages for statistical significance and reported p-values in Table 3, and included new text (in red) in the results section to describe this.
3. Figure 2 and Figure 3 do not present units of measure on the Y-axis.
With all due respect, we are not entirely sure what this comment is referring to as both Figures 2 and 3 do include units of measure on the Y-axis. Perhaps we are misunderstanding something? Thank you for clarifying this.
4. In Figure 2 it is not clear if the p-value and Cohen's d is the difference between the groups or between the pre and post-TREK? (The figure should be self-explanatory).
Significance values and effect sizes represent pre-post changes within each respective group, and not comparisons between survivors and caregivers. We have included this guidance as a note in Figure 2.
5. In Figure 3, the hardest and most interesting data of the study, it is convenient to place error bars.
We have placed error bars in Figure 2 as suggested.
6. In the discussion, it is possible that the decrease in IL-6 and CRP concur, as explained by the authors. Would this result be the expected one to replicate the study or a similar investigation? If this is the case, the possibility of a distinctive mechanism of concurrent inflammatory activity should be suggested for the practice of Mindfulness.
We have included language about this in the discussion.
Reviewer 2 Report
This is a well-done analysis addressing an important problem. I just had a few minor comments:
- In some parts of the paper, I had difficulty figuring out if the caregivers are also cancer survivors. Revisions of the text in the relevant parts of the paper is necessary to prevent this confusion.
- Table 1 and Table 2 needs more detailed captions, in particular, to explain some of the gaps.
Author Response
We thank Reviewer 2 for their helpful suggestions. Please find our responses under your suggestion.
- In some parts of the paper, I had difficulty figuring out if the caregivers are also cancer survivors. Revisions of the text in the relevant parts of the paper is necessary to prevent this confusion.
We apologize for this confusion. We have added clarifying language in the Participants and Enrollment section to state that either young adult cancer survivors or adult caregivers were enrolled but not both caregivers who were also survivors.
- Table 1 and Table 2 needs more detailed captions, in particular, to explain some of the gaps.
We have added language as a note in both tables to clarify this.